# Comparing Thixotropic and Herschel-Bulkley Parameterizations for Continuum Models of Avalanches and Subaqueous Debris Flows

Chan-Hoo Jeon[1,2] and Ben R. Hodges[2]

[1]Division of Marine Science, The University of Southern Mississippi, 1020 Balch Blvd, Stennis Space Center, Mississippi 39529, USA
[2]Center for Water and the Environment, The University of Texas at Austin, Austin, Texas 78712, USA

*Correspondence to:* Chan-Hoo Jeon (chanhoo.jeon@usm.edu)

**Abstract.** Avalanches and subaqueous debris flows are two cases of a wide range of natural hazards that have been previously modeled with non-Newtonian fluid mechanics approximating the interplay of forces associated with gravity flows of granular and solid-liquid mixtures. The complex behaviors of such flows at unsteady flow initiation (i.e. destruction of structural jamming) and flow stalling (restructuralization) imply that the representative viscosity-stress relationships should include hysteresis: there is no reason to expect the time scale of microstructure destruction is the same as the time scale of restructuralization. The non-Newtonian Herschel-Bulkley relationship that has been previously used in such models implies complete reversibility of the stress-strain relationship and thus cannot correctly represent unsteady phases. In contrast, a thixotropic non-Newtonian model allows representation of initial structural jamming and aging effects that provide hysteresis in the stress-strain relationship. In this study, a thixotropic model and a Herschel-Bulkley model are compared to each other and to prior laboratory experiments that are representative of an avalanche and a subaqueous debris flow. A numerical solver using a multi-material level-set method is applied to track multiple interfaces simultaneously in the simulations. The numerical results are validated with analytical solutions and available experimental data using parameters selected based on the experimental setup and without post-hoc calibration. The thixotropic (time-dependent) fluid model shows reasonable agreement with all the experimental data. For most of the experimental conditions, the Herschel-Bulkley (time-independent) model results were similar to the thixotropic model, a critical exception being conditions with a high yield stress where the Herschel-Bulkley model did not initiate flow. These results indicate that the thixotropic relationship is promising for modeling unsteady phases of debris flows and avalanches, but there is a need for better understanding of the correct material parameters and parameters for the initial structural jamming and characteristic time of aging, which requires more detailed experimental data than presently available.

# 1 Introduction

A wide range of natural hazards involve gravity-driven flows down a slope, e.g. landslides (terrestrial or submarine), flood-driven debris flows, mudflows, lahars, avalanches, and volcanic lava flows. Such flows range from relatively homogeneous particles (e.g. snow avalanches) to extremely heterogeneous (terrestrial landslides) and generally can be classified by solid concentration, material type, and mean velocity (Pierson and Costa, 1987; Smith and Lowe, 1991; Coussot and Meunier, 1996; Locat and Lee, 2002). Avalanches (e.g. snow, rock) are typically considered dry granular flows, whereas debris flows are liquid/solid mixtures where the solids are a dominant forcing, which can be contrasted to flood flows where sediment solids play a secondary role (Iverson, 1997). In theory, avalanche flows at the homogeneous end of the spectrum should be amenable to direct modeling as particles (granular flows), although it remains to be seen whether sufficient computer power can ever be practically applied for large-scale natural hazards. Flows with heterogeneous mixtures of liquids and solids provide further challenges as we simply do not have an adequate and proven theory for representing their behavior at natural hazard scales. Indeed, even if we develop a complete and practical theory for the movement of a mixture of fluid, particles, and entrained large objects across several magnitudes of scales, it is unclear how we would effectively capture the uncertainty associated with size and space distribution of solid objects (e.g. boulders in a landslide) that affect the flow propagation in any model attempting to directly represent fluid-solid structural interactions.

Large-scale natural hazard flows have been widely investigated with field observations, small-scale laboratory experiments, and numerical models. A common observation is that the complexity of the material composition and the effective rheological characteristics play important roles in material movement (Malet et al., 2003; Bisantino et al., 2010; Jeong, 2014; de Haas et al., 2015). This flow complexity is illustrated by the classification of subaqueous mass movements by Locat and Lee (2002) into five types with different behaviors: slides, topples, spreads, falls, and flows. At the "flow" end of the spectrum the water content is high, the particle sizes are small, and the flowing conditions are reasonably considered a fluid continuum. As the water content decreases and/or the particle size distribution covers more orders of magnitude, the theoretical basis for the fluid continuum approach becomes weaker and requires more empirical parameterization to capture other behaviors. Furthermore, the transition between a non-moving to a flowing regime can involve spatial heterogeneity and time-dependent behavior that is not well-represented by parameterizations of the flowing regime. Real-world debris flows include additional complexity as they erode and entrain material along the bottom and sides of the slope with the downstream flow. We take these issues as motivational for the present work and refer the reader to the recent review of Delannay et al. (2017) for further insight on granular flows and Shanmugam (2015) for heterogenous flows. The fundamentals physics of such flows is presented in Iverson (1997). Herein, we do not seek to distinguish between the differing physics of these various complex flows, but focus on advancing the use of non-Newtonian viscosity models as a proxy for their general behavior. For simplicity in exposition, we will use the term "debris flow" to refer to any real-world mixture modeled as a continuum fluid using a non-Newtonian model.

Following Ancey (2007), the existing approaches to simulating debris flows can be categorized in three groups: (i) applying soil mechanics concept of Coulomb behavior, which provides reasonable solutions for heterogeneous granular mass flows (Iverson and Denlinger, 2001; Iverson, 2003), (ii) merging soil and fluid mechanics models, and (iii) representing the

heterogeneous debris as a continuum fluid with behaviors similar to a non-Newtonian fluid (the approach herein) where the transition from a stable structure to a moving fluid is handled as a viscous effect. This is not to imply that such flows are actually non-Newtonian fluids, merely that some of their behaviors can be captured with an appropriately parameterized viscosity model (e.g. Davies, 1986; Pierson and Costa, 1987; Coussot and Meunier, 1996; Pudasaini, 2012). Indeed, Iverson (2003) has

referred to the rheological approach to debris flows as a "myth" and argued for its replacement with mixture models using separate solid-fluid components. However, their argument remains contentious and it is not clear that the present state of mixture models is substantially less mythical than application of a rheological model when considering heterogenous mixtures over a wide range of scales. Given that debris flow covers such diverse phenomena and complex physics, it seems likely the "correct" model for the foreseeable future will be the model that best fits a specific event, experiment, or flow type of interest.

In the absence of research that definitively solves the conundrum of debris flow, we follow the long history of using rheological models as a proxy. Such models are parsimonious in the number of coefficients and are effectively agnostic to the inherent uncertainties of fluid-solid distributions and interactions. In using a non-Newtonian rheological model, the real-world interaction between solid particles and surrounding fluid in a heterogeneous mixture can be thought of as similar to the microstructural behavior of a homogeneous non-Newtonian fluid where the local fluid viscosity is a function of the local stress. The main

advantage of this approach is that a non-Newtonian rheological model is simply a time/space dependent viscosity term for the Navier-Stokes equations. It follows that the time/space-varying eddy viscosities in a wide range of existing hydrodynamic codes can be readily adapted to non-Newtonian behavior and used for parameterized modeling of debris flows.

Note that the terminology of non-Newtonian flows can be confusing as a "time-independent" models have viscosities that can change with both space and time throughout a flow. The difference between a "time-independent" and a "time-dependent" non-

Newtonian fluid is whether the relation between stress and viscosity (i.e. non-Newtonian equation itself) is allowed to change with time. Thixotropic (time-dependent) fluids are defined as non-Newtonian fluids where the process of "aging" during a flow changes the underlying fluid microstructure and the relationship between stress and viscosity (Moller et al., 2009). Herein, we examine how the use of a thixotropic model provides the ability to model behaviors that cannot be represented with a time-independent non-Newtonian model. Our goal is to provide insight into the research needs for further experiments and model

development into the natural hazards of gravity-driven debris flows across the transitions from inception to stalling.

Gravity-driven debris flows have a range of triggering mechanisms and their composition evolves from initiation through motion and deposition or stalling, which can include a variety of behaviors that make modeling a challenge (Iverson, 1997). Parameterized non-Newtonian fluid models are an obvious approach to approximate these behaviors. Time-independent rheological models have been widely used to simulate debris flows (e.g. Bovet et al., 2010; Pirulli, 2010; Tsai et al., 2011; Manga

and Bonini, 2012), however the real-world flow characteristics include time-dependent behaviors that could be categorized as "thixotropic" (Perret et al., 1996; Crosta and Dal Negro, 2003; Bagdassarov and Pinkerton, 2004; Aziz et al., 2010). Our focus in this paper is examining how a thixotropic model behavior compares to the more common time-independent (Herschel-Bulkley) non-Newtonian fluid model.

From a macroscale perspective, debris flows have similar behaviors to "yield-stress fluids" that have been studied as a class of

non-Newtonian fluids (Møller et al., 2006; Scotto di Santolo et al., 2010). A yield-stress fluid is effectively a solid (i.e. infinite

viscosity) below a critical stress value (yield stress). This behavior is similar to what might be expected from a debris mixture of liquid and solids that is initially at rest and is triggered into motion as the yield stress is exceeded, which is the basis for prior time-independent non-Newtonian models cited above. At the microscale under low stress (near rest) conditions the fluid flow around the solids in a debris mixture is inhibited by viscous boundary layers and inertia of the solids, which provides effects similar to a higher-viscosity fluid at the macroscale (i.e. low deformation under stress). Once the solids in the debris have accelerated the effects of particle lift, drag, and rotation induced by the surrounding turbulent fluid flow, as well as solid-solid impacts and particle disintegration, will provide behaviors similar to a lower-viscosity fluid that deforms more easily under stress. This change from high viscosity to low viscosity under stress is readily simulated with a conventional Herschel-Bulkley, time-independent, non-Newtonian model. Arguably, what is missing from a time-independent model is that the destruction of the initial microstructure of the debris can change the effective macroscale viscosity and response to stress. If the flow stalls either globally or locally, it may take some time to re-establish its microstructure so the yield stress for a recently-stalled flow should be different than the yield stress after aging (consolidation). We can think of the behavior of a debris flow as controlled, at least partly, by the evolution of the microstructure and requiring a time-dependent element in the non-Newtonian model.

The simplest non-Newtonian yield-stress fluids are Bingham plastics. More complex behaviors are associated with "shear thinning" and "shear thickening" where the effective viscosity nonlinearly changes with the rate-of-strain. For these standard cases, the relationship between viscosity and rate-of-strain is repeatable and time-independent. The approach proposed by Herschel and Bulkley (1926) is a common approach for representing the general case of time-independent non-Newtonian fluids wherein the plastic viscosity, $\eta$, is conditional on the yield stress, $\tau_0$, as

$$
\begin{cases}
\eta = K\dot{\gamma}^{n-1} + \frac{\tau_0}{\dot{\gamma}} & \text{if } \tau > \tau_0 \\
\dot{\gamma} = 0 & \text{if } \tau \leq \tau_0
\end{cases}
\tag{1}
$$

where $K$ is the consistency parameter, $n$ is the Herschel-Bulkley fluid index, and $\dot{\gamma}$ is the scalar value of the rate of strain. The Herschel-Bulkley fluid index $n$ controls the overall modeled behavior, where $0 < n < 1$ is shear thinning, $n > 1$ is shear thickening and $n = 1$ corresponds to the Bingham plastic model (Bingham, 1916).

A recognized problem with numerical simulation using a Herschel-Bulkley model is the viscosity is effectively infinite below the yield stress, i.e. the condition $\dot{\gamma} = 0$ in Eq. (1) is identical to $\eta = \infty$ for modeling a fluid continuum that becomes solid below the yield stress. An infinite (or even very large) viscosity creates an ill-conditioned matrix in a discrete solution of the partial differential equations for fluid flow. Furthermore, the instantaneous transition from infinite to finite viscosity as the yield stress is crossed provides a sharp change that can lead to unstable numerical oscillations. Dent and Lang (1983) attempted to resolve this issue with a bi-viscous Bingham fluid model for computing motion of snow avalanches. Their approach was shown to be reasonable using comparisons with experimental data, but was later determined to be invalid for conditions where the shear stresses are much lower than the yield stress (Beverly and Tanner, 1992). A more successful approach was that of Papanastasiou (1987), who proposed modifying the Herschel-Bulkley model with an exponential parameter, $m$. The Papanastasiou model (presented in detail in Section 3, below), with appropriate values for $m$, shows good approximations at low shear rates for Bingham plastics (Beverly and Tanner, 1992).

Although a flow simulated with the Papanastasiou model will have changes in the viscosity with time (as the shear changes with time), the model is still deemed "time-independent" as the relationship between viscosity and shear is fixed by the selection of $K$, $n$, $m$, and $\tau_0$. Arguably, there exists a wide range of debris flows over which the Papanastasiou approach should be adequate, as the *time-dependent* characteristics of debris flows are, at least theoretically, principally confined to the initiation and cessation of the flow, i.e. when the microstructure of the debris is evolving and changing the relationship between shear and viscosity. It follows that steady-state conditions for debris flows should be reasonably represented with time-independent models. Indeed, O'Brien and Julien (1988) concluded, by their experiments, that mud flows whose volumetric sand concentration were less than 20% showed the behavior of a silt-clay mixture, which can be described reasonably well by the Bingham plastic model at low shear rates and a time-independent Herschel-Bulkley model at high shear rates. Liu and Mei (1989) reported good agreement for theory and experiment with a Bingham plastic model and a homogeneous mud flow that provides a steady front propagation speed (necessarily long after the initiation phase). The Herschel-Bulkley model has also been used to simulate debris flow along a slope, but reported results have discrepancies with experimental data, especially in the early stages (Ancey and Cochard, 2009; Balmforth et al., 2007). Bovet et al. (2010) applied the time-independent Papanastasiou model to simulate snow avalanches with some success, but again their results showed more significant discrepancies with experiments during flow initiation. De Blasio et al. (2004) simulated both subaerial and subaqueous debris flows with a Bingham fluid model. Their results for the subaerial debris flows were in a reasonable agreement with laboratory data, but their subaqueous simulations showed a significant discrepancy with measurements. A clear challenge in validating models of debris flows beyond steady conditions is that the most commonly available experimental data is focused on the steady or quasi-steady conditions after the debris structure has (relatively) homogenized.

Thixotropic (time-dependent) behavior, which is not represented in the Herschel-Bulkley model, provides an interesting avenue for representing the expected macro-scale behavior of a debris flow near initiation. At rest, debris solids provide structural resistance to flow (for denser solids), and a greater inertial resistance to motion than the fluid. Thus, it is reasonable to expect initial behavior similar to a Bingham plastic; i.e. initially-infinite viscosity with a high yield stress. However, the onset of motion for the debris flow begins the destruction of the microstructure, homogenization of the debris, and a change in the relationship between stress and viscosity, which might be thought of as shear-thinning behavior. A key difference between a Herschel-Bulkley model and the real world is that the former requires a return to structure whenever the internal stress drops below the yield stress, however, in a debris flow we expect the destruction of microstructure to significantly reduce the stress at which renewal of structure (consolidation) occurs. For a real debris flow we expect different viscosity-stress behaviors during initiation, steady-state, and slowing phases (consistent with evolving microstructure), but a time-independent Herschel-Bulkley model is effectively an assumption that the processes of destruction of microstructure and renewal are exactly reversible. For a thixotropic fluid the time dependency can occur as part of spatial gradients that evolve over time; e.g. high shear stress is localized in a small region by heterogeneity of particles, and in this region the fluid begins to yield (Pignon et al., 1996). Thus, in a thixotropic fluid there is spatial-temporal destruction of microstructure that leads to changes in the effective viscosity that cannot be represented in the standard time-independent models. Coussot et al. (2002a) proposed an empirical viscosity model for thixotropic fluids (presented in detail in Section 3, below), which captures these fundamental behaviors.

Prior research on thixotropic flows has mainly focused on laboratory experiments (Mohrig et al., 1999; Chanson et al., 2006; Sawyer et al., 2012; Haza et al., 2013), although a few studies have numerically investigated the characteristics of thixotropic flow on a simple inclined plane (Huynh et al., 2005; Hewitt and Balmforth, 2013). In general, numerical simulation results have not been well validated by the experimental data, arguably due to limitations in both non-Newtonian viscosity models and the sparsity of available laboratory data. Thixotropic flows modeled at the laboratory scale typically use clays (e.g. Bentonite, Kaolin) to create the microstructure controlling non-Newtonian behavior (Balmforth and Craster, 2001). Preparation of a homogenous clay suspension for such experiments is a demanding task, the details of which can be found in Coussot et al. (2002b), Huynh et al. (2005), and Chanson et al. (2006). Unfortunately, we cannot expect the structure of a heterogeneous large-scale debris flow to mimic the flow scales, yield stresses, and parameters for a homogeneous thixotropic laboratory flow. However, lacking data from a large-scale debris flow that could be adequately used for model comparisons, herein we take a first step by analyzing how thixotropic models compare to time-independent models for laboratory-scale flows.

Validating the use of a non-Newtonian model to represent a real-world debris flow presents challenges on two levels: first, does the model correctly represent a non-Newtonian flow? Second, does the non-Newtonian flow (when parameterized) represent a real-world debris flow? To date, successful non-Newtonian models of real-world flows have been parameterized using a time-independent approach, which limits the ability of the model to represent the transition phases, i.e. flow initiation and stalling (e.g. Bovet et al., 2010; Pirulli, 2010; Tsai et al., 2011; Manga and Bonini, 2012). Unfortunately, data on transition phases for real-world flows is lacking, and is severely limited even for laboratory-scale flows.

In this paper we evaluate a time-independent Papanastasiou model and a time-dependent Coussot model for simulations of laboratory-scale avalanche and subaqueous debris flows, with comparisons to available experimental measurements. The governing equations are presented in Section 2 and the non-Newtonian Papanastasiou and Coussot viscosity models in Section 3. A key confounding issue for model/experiment comparisons is the estimation of parameters for a non-Newtonian fluid model (in particular the initial degree of jamming), which we discuss in Section 4. The numerical solver, using a multi-material level-set method, is presented in Section 5. The solver is validated in Section 6 with the analytical solutions for the Poiseuille flow of a Bingham fluid. In Section 7 the solver is used to model a laboratory flow that is a reasonable proxy of a thixotropic avalanche. In Section 8 we present the numerical simulations of subaqueous debris flows with three interfaces: debris-water, debris-air, and water-air, and compare our results to prior experimental data. We discuss the results and summarize conclusions in Section 9.

## 2   Governing Equations:

The governing equations in conservation form for unsteady and incompressible fluid flow can be written as (Ferziger and Perić, 2002)

$$\nabla \cdot \mathbf{u} = 0 \tag{2}$$

$$\frac{\partial \mathbf{u}}{\partial t} + \nabla \cdot (\mathbf{u} \otimes \mathbf{u}) = \frac{1}{\rho}\Big(-\nabla p + \nabla \cdot \mathbf{T} + \mathbf{f}\Big) \tag{3}$$

where $\mathbf{u}$ is the velocity vector, $\rho$ is the density, $p$ is the pressure, $\mathbf{f}$ includes additional forces such as gravitational force, surface tension force, and Coriolis force, $\mathbf{u} \otimes \mathbf{u}$ is the dyadic product of the velocity vector $\mathbf{u}$, and $\mathbf{T}$ is the viscous stress tensor:

$$\mathbf{T} = 2\eta\mathbf{D} \tag{4}$$

where $\eta$ denotes the plastic viscosity and $\mathbf{D}$ is the rate of strain (deformation) tensor:

$$\mathbf{D} = \frac{1}{2}\big[\nabla\mathbf{u} + (\nabla\mathbf{u})^T\big]. \tag{5}$$

where the superscript $T$ indicates a matrix transpose. The $\eta$ in the above is constant in time and uniform in space for a Newtonian fluid, but is potentially some nonlinear function of other flow variables for a non-Newtonian fluid.

The non-Newtonian fluid models herein use the local velocity rate-of-strain to update the plastic viscosity, $\eta$, as shown in Section 3, which makes the approach compatible with a wide range of numerical solvers that include a time/space-varying
eddy viscosity.

Equations (2) and (3) can be integrated over a control volume and, by applying the Gauss divergence theorem, we obtain the basis for the common finite-volume numerical discretization (Ferziger and Perić, 2002). For simplicity in the present work, we limit ourselves to a two-dimensional (2D) flow field for a downslope flow and the orthogonal (near-vertical) axis, which effectively assumes uniform flow in the cross-stream axis. The external force term $\mathbf{f}$ represents the gravitational force only,
neglecting surface tension forces and Coriolis. The advection term is discretized with the fifth-order WENO (Weighted Essentially Non-Oscillatory) scheme (Shi et al., 2002) or the second-order TVD (Total Variation Diminishing) Superbee scheme (Darwish and Moukalled, 2003) in separate numerical tests. The diffusion term on the right hand side of Eq. (3) is discretized with the second-order central differencing scheme. The time derivative term for the momentum equations is integrated by the second-order Crank-Nicolson implicit scheme. The deferred correction scheme (Ferziger and Perić, 2002) is applied and ghost
nodes are evaluated by the Richardson extrapolation method for high accuracy at the boundaries. The pressure gradient term is calculated explicitly and then corrected by the first-order incremental projection method (Guermond et al., 2006). To evaluate the values at the cell surfaces, the Green-Gauss method is used and the momentum interpolation scheme (Murthy and Mathur, 1997) is applied. The code is parallelized with MPI (Message Passing Interface), and PETSc (Portable, Extensible Toolkit for Scientific Computation) (Balay et al., 2016) is used for standard solver functions (e.g. the stabilized version of Biconjugate
Gradient Squared method with pre-conditioning by the block Jacobi method). The developed code has been verified by the method of manufactured solutions (further details provided in Jeon, 2015).

## 3  Non-Newtonian Fluid Models:

The Herschel-Bulkley model, Eq. (1), was made more practical for modeling a fluid flow continuum by Papanastasiou (1987), whose approach can be represented as :

$$\eta = \begin{cases} K\dot{\gamma}^{n-1} + \frac{\tau_0\left(1 - e^{-m\dot{\gamma}}\right)}{\dot{\gamma}} & \text{for all } \dot{\gamma} \\ K\dot{\gamma}^{n-1} + m\tau_0 & \text{as } \dot{\gamma} \to 0 \end{cases} \tag{6}$$

here $m$ has dimension of time such that as $m \to \infty$ we recover the original Herschel-Bulkley model with $\eta \to \infty$, whereas $m = 0$ is a simple Newtonian fluid. The scalar value of the rate of strain is obtained from $\dot{\gamma} = 2\sqrt{|\mathbf{II}_D|}$ where $\mathbf{II}_D$ is the second invariant of the rate of strain as (Mei, 2007)

$$\mathbf{II}_D = \frac{1}{2}\left[(\text{tr}(\mathbf{D}))^2 - \text{tr}\left(\mathbf{D}^2\right)\right] = D_{11}D_{22} - D_{12}^2 \tag{7}$$

and $D_{ij}$ denotes the $(i,j)$ component of the strain tensor $\mathbf{D}$ in Eq. (5). As with the Herschel-Bulkley model on which it is based, the Papanastasiou model is time-independent.

In contrast, the time-dependent (thixotropic) model of Coussot et al. (2002a) introduces dependency on a time-varying microstructure parameter ($\lambda$) in the general form:

$$\eta = \eta_0\left(1 + \omega\lambda^n\right) \tag{8}$$

where $\eta_0$ is the asymptotic viscosity at high shear rate, $\omega$ is a material-specific parameter, and $n$ is the Herschel-Bulkley fluid index. The microstructural parameter of the fluid, $\lambda$, is evaluated using a temporal differential equation:

$$\frac{d\lambda}{dt} = \frac{1}{T_0} - \alpha\dot{\gamma}\lambda \tag{9}$$

where $T_0$ is the characteristic time of the microstructure, $\alpha$ is a material-specific parameter, and $\dot{\gamma}$ is the rate of strain (as in the Herschel-Bulkley and Papanastasiou models, above). Here $\alpha$ represents the strength of the shear effect associated with inho-
mogeneous microstructure (Liu and Zhu, 2011). That is, larger of values of $\alpha$ require greater microstructure homogenization (smaller $\lambda$) to drive the system to steady-state conditions ($d\lambda/dt \to 0$).

## 4   Estimation of parameters for time-dependent Coussot model

The time-dependent Coussot model requires parameters for the asymptotic viscosity ($\eta_0$), Herschel-Bulkley fluid index ($n$), characteristic time ($T_0$), and two material-specific parameters ($\omega$ and $\alpha$) that control the response (destruction) of the mi-
crostructure. Additionally, an initial condition for $\lambda_0$ is required to solve the ordinary differential equation presented as Eq. (9). The parameters $\eta_0$ and $n$ are easily obtained from the time-independent Herschel-Bulkley model, which are typically available in experimental studies. However, the other parameters of the Coussot model are more troublesome.

As $\lambda$ represents the microstructure in the Coussot model, $\lambda_0$ can be thought of as the initial degree of jamming caused by the microstructure (i.e. the structure that must be broken down to create fluid flow). As yet, there does not appear to be an accepted
method to estimate $\lambda_0$. We propose two methods evaluating $\lambda_0$, and test these in the accompanying simulations. As discussed below, Method A is a simple analytical approach based on the critical stress, whereas Method B uses a graphical approach.

 – **Method A**: Assuming all other parameters of the fluid are known, including the critical stress $\tau_c$, the initial condition, $\lambda_0$, can be evaluated using the Coussot equation for the critical stress as Coussot et al. (2002a):

$$\tau_c = \frac{\eta_0\left(1 + \omega\lambda_0^n\right)}{\alpha T_0\lambda_0} \tag{10}$$

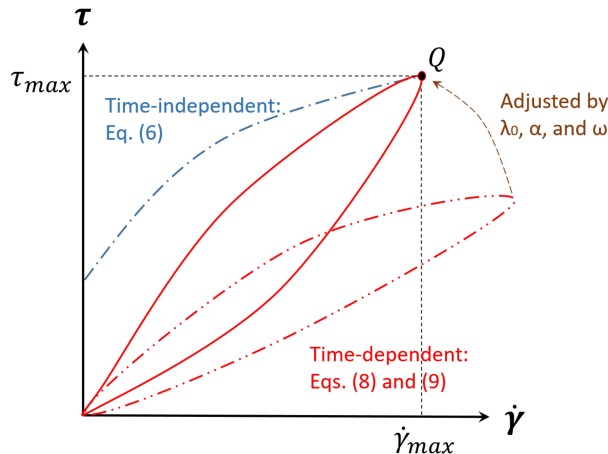

**Figure 1.** Concept of a graphical Method B for estimating a consistent set of $\lambda_0, \alpha, \omega$ parameters for Coussot model.

Unfortunately parameter values for $\omega$ and $\alpha$ also do not have well-defined estimates in the literature, so herein we adjust these to ensure real solutions for $\lambda_0$. However, in some simulations (see Section 7) this method appears to over-estimate shear stress. Furthermore, obtaining real solutions for $\lambda_0$ by perturbing $\alpha$ and $\omega$ can be time-consuming.

– **Method B**: Our second approach (which is preferred) is to approximate the critical shear stress ($\tau_c$) of a time-dependent fluid model using the maximum shear stress ($\tau_{\max}$) of a time-independent fluid model. This implies that on a graph of stress $v$. strain ($\tau : \dot{\gamma}$), the critical stress-strain point of the time-independent model should match the maximum stress point of the time-dependent model (i.e. the point where hysteresis causes the time-dependent model to operate along a different $\tau : \dot{\gamma}$ curve). This point is labeled $Q$ in Fig. 1. It is a relatively simple graphical trial and error exercise to adjust $\lambda_0$, $\omega$, and $\alpha$ to obtain the correct $Q$ for a given $T_0$, $\eta_0$, and $n$. In this approach, the most important question is how to set the matching point, $Q$. In our avalanche model (Section 7), the point $Q$ is known because the critical shear stress is given in the experimental paper. However, for our debris-flow model (Section 8), only time-independent parameters are given in the corresponding experimental report. Thus, the matching point $Q$ for this case was set where the maximum rate of strain of the thixotropic model was the same as the maximum rate of strain of the Herschel-Bulkley model.

The $T_0$ of the Coussot model in Eq. (10) can also be troublesome to estimate. This characteristic time for aging, which Coussot et al. (2002b) described as "spontaneous evolution of the microstructure," is not widely used and the literature does not provide insight on how to evaluate $T_0$ as a function of other rheological characteristics. Furthermore, $T_0$ has slightly different definitions by authors of several papers. Chanson et al. (2006) defined it as the characteristic time without any further measurement in their experiments, but provided another parameter, "rest time", used to set up the Bentonite suspensions in laboratory experiments in the result tables. However, Møller et al. (2006) defined $T_0$ as "the characteristic time of build-up of the microstructure at rest". Their characteristic time is close to the rest time of Chanson et al. (2006). Therefore, we make the assumption that the "rest time" measured in the Chanson's experiments is the same with the $T_0$ of Coussot for the

thixotropic avalanche simulations (Section 7). For simulations of subaqueous debris flow (Section 8), the experiments did not report any time scales that could be used to estimate $T_0$, so we included it as an unknown in the Method B described above. In general, graphical Method B provides a simple means to estimate a consistent set of time-dependent parameters from the time-independent parameters, which provides confidence that time-dependent and time-independent models are being compared in a reasonable manner.

## 5 Multi-material Level-Set Method:

Some types of debris flow, e.g. avalanches, can be reasonably modeled as a single fluid with a free surface where dynamics of the overlying fluid (in this example, air) are neglected. In contrast, subaqueous debris flows are more likely to require coupled modeling between lighter overlying water (Newtonian fluid) and heavier non-Newtonian debris. It is also possible to imagine more complex configurations where simultaneous solution of multiple debris layers or perhaps debris, water, and air might be necessary. For general purposes, it is convenient to apply a multi-material level-set method so that any number of fluids with differing Newtonian and non-Newtonian properties can be considered. When only two fluids are considered, the multi-material level-set method corresponds to the general level-set method for two-phase flow. The level-set method has a long history in multiphase fluids (Sussman et al., 1994; Chang et al., 1996; Sussman et al., 1998; Peng et al., 1999; Sussman and Fatemi, 1999; Bovet et al., 2010), and is based on using a $\phi_i$ distance (level set) function to represent the distance of the $i$ material (or material phase) from an interface with another material (Osher and Fedkiw, 2001).

The multi-material level-set method herein follows Merriman et al. (1994) with the addition of high-order numerical schemes (Shu and Osher, 1989; Shi et al., 2002). The "level set", of the $i$-th fluid is designated as $\phi_i$, where

$$\phi_i \equiv \begin{cases} +d_i(\mathbf{x}, \Gamma_i) & \text{if } \mathbf{x} \text{ inside } \Gamma_i \\ -d_i(\mathbf{x}, \Gamma_i) & \text{if } \mathbf{x} \text{ outside } \Gamma_i \end{cases} \tag{11}$$

where $i = \{1, 2, \cdots, N_m\}$, $N_m$ is the number of materials, $\Gamma_i$ is the interface of fluid $i$, and $d$ is the distance from the interface. The density and viscosity at a computational node for the multiple fluid system are evaluated from a combination of the individual fluid characteristics based on an approximate Heaviside function that provides a continuous transition over some $\epsilon$ distance on either side of an interface:

$$\rho \equiv \sum_{i=1}^{N_m} \rho_i H_i \quad , \quad \eta \equiv \sum_{i=1}^{N_m} \eta_i H_i \tag{12}$$

where the Heaviside function for fluid $i$ is

$$H_i(\phi_i) \equiv \begin{cases} 0 & \text{if } \phi_i < -\epsilon \\ \frac{1}{2} \left[ 1 + \frac{\phi_i}{\epsilon} + \frac{1}{\pi} \sin\left(\frac{\pi \phi_i}{\epsilon}\right) \right] & \text{if } |\phi_i| \leq \epsilon \\ 1 & \text{if } \phi_i > \epsilon \end{cases} \tag{13}$$

where $2\epsilon$ is therefore the finite thickness of the numerical interface between fluids.

The level-set initial condition is simply the distance from any grid point in the model to an initial set of interfaces, i.e. $\phi_i = d_i$. Note that each point has a distance to each $i$ interface. The level set is treated as a conservatively-advected variable that evolves according to a simple non-diffusive transport equation (Osher and Fedkiw, 2001):

$$\frac{\partial \phi_i}{\partial t} + \mathbf{u} \cdot \nabla \phi_i = 0 \tag{14}$$

The above is coupled to solution of momentum and continuity, Eqs. (2) and (3), to form a complete level-set solution for fluid flow. The continuous interface $i$ at time $t$ is located where $\phi_i(\mathbf{x}, t) = 0$. In general, the $i$ interface will be between the discrete grid points of the numerical solution, so it is found by multi-dimensional interpolation from the discrete $\phi_i$ values. After advancing the level set from $\phi(t)$ to $\phi(t + \Delta t)$, the values of the level set will no longer satisfy the Eikonal condition of $|\nabla \phi_i| = 1$; that is, the level-set values on the grid cells obtained by solving Eq. (14) are no longer equidistant from the interface (i.e. the zero level set). It is known that if the level sets are naively evolved through time without satisfying the Eikonal condition the Heaviside functions will become increasingly inaccurate (Sussman et al., 1994). This problem is addressed with "reinitialization," which resets the $\phi(t + \Delta t)$ to satisfy the Eikonal condition. The simplest approach to reinitialization is iterating an unsteady equation in pseudo-time to steady state such that the steady-state equation satisfies the Eikonal condition (Sussman et al., 1998). Let $\hat{\phi}$ be an estimate of the reinitialized value for $\phi(t + \Delta t)$ in the equation

$$\frac{\partial \hat{\phi}_i}{\partial \mathcal{T}} + \mathbf{S}(\hat{\phi}_i) \left( |\nabla \hat{\phi}_i| - 1 \right) = 0 \tag{15}$$

where $\mathcal{T}$ is the pseudo time, and $\mathbf{S}$ is the signed function as (Sussman et al., 1998)

$$\mathbf{S}(\hat{\phi}_i) = \begin{cases} -1 & \text{if} \quad \hat{\phi}_i < 0 \\ 0 & \text{if} \quad \hat{\phi}_i = 0 \\ 1 & \text{if} \quad \hat{\phi}_i > 0 \end{cases} \tag{16}$$

The time-advanced set of $\phi(t + \Delta t)$ is the starting guess for $\hat{\phi}$, and the steady-state solution of $\hat{\phi}$ will satisfy $|\nabla \hat{\phi}_i| = 1$ to numerical precision.

For the present work, the advection term in Eq. (14) is discretized with the fifth-order WENO scheme, and the time derivative term is integrated by the third-order TVD Runge-Kutta method (Shu and Osher, 1989). For the reinitialization step of Eq. (15), the second-order ENO (Essentially Non-Oscillatory) scheme (Sussman et al., 1998) and a smoothing approach (Peng et al., 1999) are used for the spatial discretization (further details are provided in Jeon, 2015).

## 6   Poiseuille Flow of Bingham Fluid:

A two-dimensional Poiseuille flow in a channel driven by a steady pressure gradient of $\partial p/\partial x$ provides a validation case for the non-Newtonian fluid solver. If gravity is considered negligible and the flow is approximated as symmetric about a centerline

**Table 1.** Bingham fluid Herschel-Bulkley model parameters used in Poiseulille flow test cases, from Filali et al. (2013)

| Term | Value |
|---|---|
| Herschel-Bulkley index ($n$) | 1.0 |
| Yield stress ($\tau_0$, Pa) | 4.0 |
| Consistency parameter ($K$, Pa·s$^n$) | 2.9 |

between two walls, then the analytical solution for the flow on one side of the centerline is (Papanastasiou, 1987):

$$
u(y) = \begin{cases}
\frac{1}{2\eta}\left(-\frac{\partial p}{\partial x}\right)\left(F^2 - y^2\right) - \left(\frac{\tau_0}{\eta}\right)(F - y) \\
\qquad \text{for} \quad F_D \leq y \leq F \\
\frac{1}{2\eta}\left(-\frac{\partial p}{\partial x}\right)\left(F^2 - F_D^2\right) - \left(\frac{\tau_0}{\eta}\right)(F - F_D) \\
\qquad \text{for} \quad 0 \leq y < F_D
\end{cases}
\tag{17}
$$

where $F$ is the distance from the center to a channel wall, $y$ is the Cartesian axis normal to the flow direction with $y = 0$ at the centerline of the flow between the two walls, $\tau_0$ is the yield stress, and $F_D$ is a length scale representing the relationship

between yield stress and the pressure gradient:

$$
F_D = \frac{\tau_0}{\left(-\frac{\partial p}{\partial x}\right)}
$$

A convenient set of Bingham fluid parameters for the Poiseuille test cases can be extracted from the dip coating study of Filali et al. (2013) as shown in Table 1. In the simulations, the distance from the centerline to a side wall is 0.05 m. Our model grid uses 320 cells in the flow direction and 32 cells in the cross-stream direction. A Neumann boundary condition is applied

along the lower boundary of the simulation domain, so the simulation includes only the upper half-channel of this symmetric flow.

Using the Papanastasiou model of Eq. (6) to approximate a Herschel-Bulkley model of a Bingham fluid requires time-scale parameter $m$ to provide smooth behavior across the yield stress threshold. We tested values of $m = \{100, 400\}$ s. As shown in Fig. 2, the numerical results are in very good agreement with the analytical solutions for both values. For this simulation, the

lower value of $m = 100$ s is reasonable for a Papanastasiou model.

## 7 Thixotropic Avalanches:

An avalanche is a granular flow of an initially-solid field that is triggered from rest into a down-slope flow. A thixotropic model of an avalanche as a fluid continuum can represent a rapid progression from local to global release of the initial structural jamming, $\lambda_0$. Chanson et al. (2006) developed dam-break experiments with a thixotropic fluid that provide reasonable facsimiles of avalanche flows if the time scale to remove the dam is smaller than the time scale for release of structural jamming. The

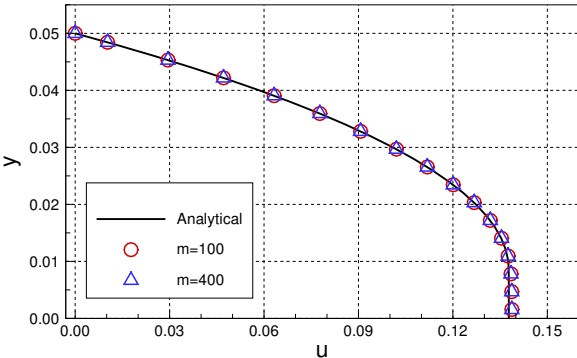

**Figure 2.** Comparison of analytical and numerical solutions for steady-state fluid velocity for Poiseuille flow of a Bingham fluid.

initial conditions of the Chanson experiments are shown in Fig. 3 where $\theta$, $d_0$, and $l_0$ represent the angle of a slope, the height of the initial avalanche that is normal to the slope, and the length of the avalanche along a slope, respectively. We modeled this same setup with our multi-material level-set solver.

The Chanson experiments identified four thixotropic flow types that were functions of the relative effect of initial structural
jamming. Weak jamming (i.e. small $\lambda_0$) characterizes Type I, such that inertial effects dominate the downstream flow (highest Re) and the flow only ceases when it reaches the experiment outfall. It follows that Type I is effectively a model of an avalanche that propagates until it is stopped by an obstacle or change in slope. Type II flows had intermediate initial jamming, which showed rapid initial flow followed by deceleration until "restructuralization" that effectively stops the downstream progression. Type II is a model of an avalanche that dissipates itself on the slope. The Type III flows, with the highest $\lambda_0$, have complicated
behavior with separation into identifiable packets of mass (typically two, but sometimes more) with different velocities. Type IV behavior was the extremum of zero flow. Chanson reported 28 experiments in total, but data on wave front propagation was provided for only five experiments (Fig. 6 in Ref. Chanson et al. (2006)) of Type I and II behavior. We simulated three of these experiments that covered a wide range of characteristics and behaviors, as shown in Table 2. Note that Chanson et al. (2006) used $\tau_{c2}$ to designate the critical shear stress during unloading (restructuralization), which we consider an approximation for
the yield stress, $\tau_0$, for a time-independent model.

We simulate the three cases of Table 2 with the time-independent Papanastasiou model of Eq. (6) and the Coussot time-dependent model of Eqs. (8) and (9). For a Bingham time-independent model, we use $n = 1$ with $K = \eta_0$ from the Chanson experiments. The smoothing value of $m = 100$ was selected based on the Poiseuille flow modeled in Section 6, above. For a Herschel-Bulkley time-independent model, we use the same $K$ and $m$ as the Bingham plastic model, but with $n = 1.1$ that was
used in the detailed technical report on the same experiments by Chanson et al. (2004). The time-dependent model requires specification of parameters $\{n, T_0, \alpha, \eta_0, \lambda_0, \omega\}$ as discussed in Section 4. The Herschel-Bulkley index in the time-dependent model uses the same value ($n = 1.1$) as the time-independent model. Two sets of values for $\{\alpha, \lambda_0, \omega\}$ are determined by the two methods (A and B) outlined in Section 4, above. Method A uses Eq. (10), which requires a value for $\tau_c$; herein this is taken

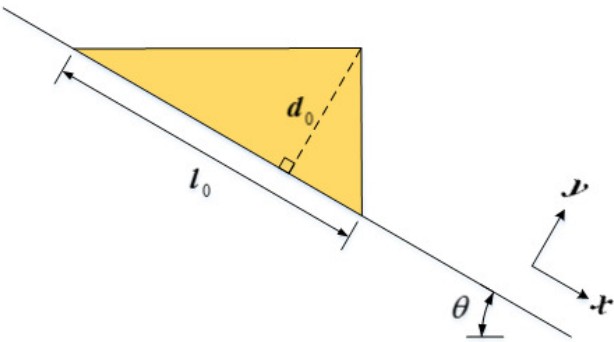

**Figure 3.** Definition sketch for initial conditions of an avalanche along a slope.

**Table 2.** Dimensions and data for thixotropic avalanche simulations corresponding with experiments by Chanson et al. (2006)

| Term | Case 1 | Case 2 | Case 3 |
|---|---|---|---|
| Chanson Experiment No. | 6 | 19 | 23 |
| Thixotropic flow Type | II | II | I |
| Slope angle ($\theta$, °) | 15 | 15 | 15 |
| Initial height ($d_0$, m) | 0.0727 | 0.0756 | 0.0732 |
| Initial length ($l_0$, m) | 0.2908 | 0.3024 | 0.2928 |
| Herschel-Bulkley index ($n$) | 1.1 | 1.1 | 1.1 |
| Yield stress (unloading, $\tau_0$, Pa) | 31.0 | 21.1 | 14.0 |
| Critical stress (loading, $\tau_c$, Pa) | 90 | 165 | 50 |
| Asymptotic viscosity ($\eta_0$, Pa$\cdot$s) | 0.062 | 0.635 | 0.555 |
| Density ($\rho$, kg/m$^3$) | 1099.8 | 1085.1 | 1085.1 |
| Characteristic (rest) time ($T_0$, s ) | 300 | 900 | 60 |

as Chanson's critical loading stress ($\tau_{c1}$ in Chanson et al., 2006) during the initial structural breakdown. Similarly, Method B requires a $\tau_{\max}$ for point Q in Fig. 1, which is also set to the critical loading stress.

For all simulations, the no-slip wall condition is applied to the bottom wall, and the number of computational cells is $512 \times 80$. The computational domain is rotated so the $x$ axis is along the sloping bed, which means that computational cell faces are either orthogonal or parallel to the slope. The gravitational constant ($g = 9.81 \, \mathrm{m \cdot s^{-2}}$) is divided into two components of ($g\sin\theta, -g\cos\theta$). The density and viscosity of air are $1.0 \, \mathrm{kg \cdot m^{-3}}$ and $1.0\mathrm{E}\text{-}5 \, \mathrm{Pa \cdot s}$, respectively.

The analytical relationships between shear stress and rate of strain for the different viscosity models are presented in Figs. 4 through 6. In these figures, "Herschel-Bulkley" and "Bingham" lines are the results of Eq. (6) with $n = 1.1$ and $n = 1.0$, respectively. The "case A" and "case B" lines denote results of Methods A and B from Section 4 for determining time-dependent parameters with Eqs. (8) and (9). The estimated parameters of $\lambda_0$, $\omega$, and $\alpha$ by two methods that are used in

**Table 3.** Parameters of the time-dependent fluid model for thixotropic avalanche simulations using Method A and Method B for setting values.

| Term | Case | | | | | |
| --- | --- | --- | --- | --- | --- | --- |
| | 1A | 1B | 2A | 2B | 3A | 3B |
| Flow index ($\omega$) | 1.0 | 0.7 | 0.5 | 1.0 | 0.1 | 1.0 |
| Material parameter ($\alpha$) | 5.67E-6 | 1.0E-6 | 3.56E-6 | 1.0E-6 | 5.33E-6 | 1.0E-5 |
| Microstructural parameter ($\lambda_0$) | 0.6631 | 0.95 | 3.8576 | 0.74 | 5.9235 | 0.29 |

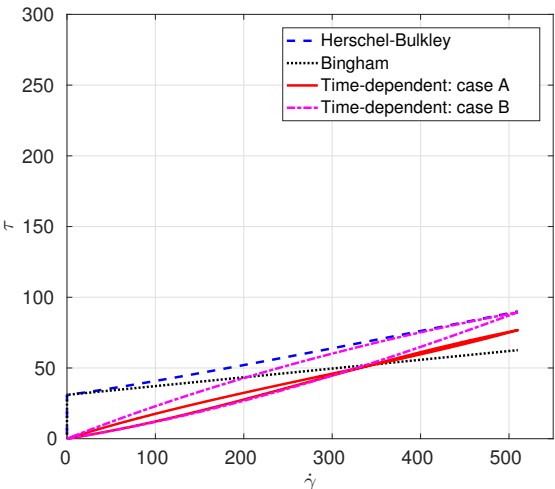

**Figure 4.** Analytical stress-strain for thixotropic avalanche Case 1: shear stress (Pa) and rate of strain (s$^{-1}$) with $\tau_0 = 31$ Pa and $\tau_c = 90$ Pa. The $\tau$ axis is scaled for comparison with Figs. 5 and 6 while the $\dot{\gamma}$ axis has an individual scale for clarity.

these figures are shown in Table 3. These results illustrate the challenge of using Method A (the critical stress relationship) for estimating $\lambda_0$. The numerical solutions of the Coussot model ordinary differential, Eq. (9), are obtained by the Runge-Kutta 4th-order method. The resulting time-dependent stress-strain relationship can be far from the time-independent relationship that is otherwise thought to be a reasonable model.

5    Propagation of the fluid wave front provides a simple means of directly comparing the temporal and spatial evolution of the model and experiments. To facilitate comparisons across experimental scales, the non-dimensionalized front location and simulation time after gate opening are $x^* = x/d_0$ and $t^* = t\sqrt{g/d_0}$, respectively. A simple theoretical estimate for the wave front propagation suitable for short time scales was derived from equations of motion as Eq. (26) in Chanson et al. (2006), repeated here as:

10  $$x_s^* = \frac{\sin\theta}{2} (t^*)^2 \tag{18}$$

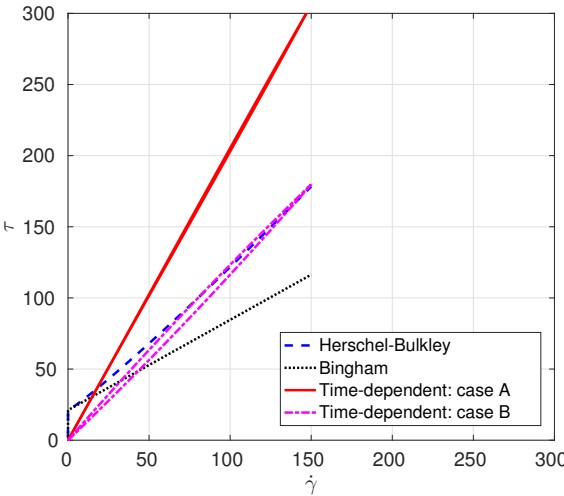

**Figure 5.** Analytical stress-strain for thixotropic avalanche Case 2: shear stress (Pa) and rate of strain ($\mathrm{s}^{-1}$) with $\tau_0 = 21.1$ Pa and $\tau_c = 165$ Pa. The $\tau$ axis is scaled for comparison with Figs. 4 and 6 while the $\dot\gamma$ axis has an individual scale for clarity.

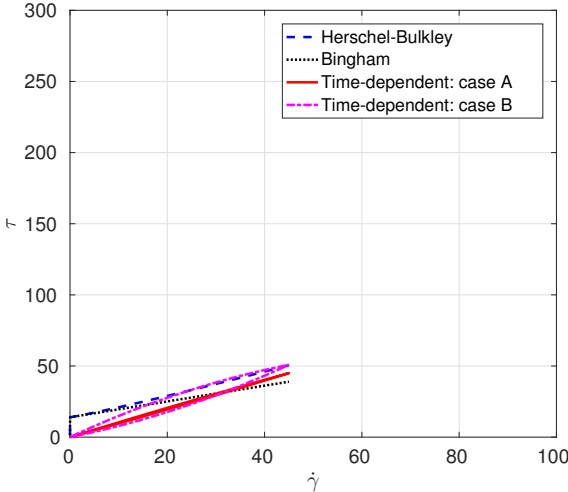

**Figure 6.** Analytical stress-strain for thixotropic avalanche Case 3: shear stress (Pa) and rate of strain ($\mathrm{s}^{-1}$) with $\tau_0 = 14$ Pa and $\tau_c = 50$ Pa. The $\tau$ axis is scaled for comparison with Figs. 4 and 5 while the $\dot\gamma$ axis has an individual scale for clarity.

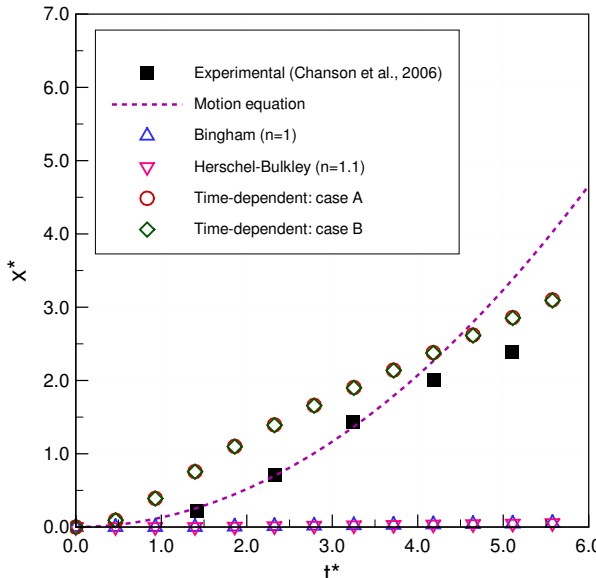

**Figure 7.** Thixotropic avalanche Case 1: comparison of numerical results and experimental data for non-dimensional front displacement ($x^*$) as a function of non-dimensional simulation time ($t^*$)

The simulation, experiment, and theory results are shown in Figs. 7, 8, and 9 for Cases 1, 2, and 3 of Table 2, respectively. The dashed line represents the theoretical solution for the propagating the front of Eq. (18).

    The most striking feature in the results is that the simulations for Cases 2 and 3 (smaller $\tau_0$) are relatively similar for all the models, whereas the time-independent models (Bingham and Herschel-Bulkley) completely fail for Case 1 (larger $\tau_0$) even though the time-dependent models continue to perform reasonably well. The failure appears to be due to an inability of the time-independent models in Case 1 to develop sufficient strain to move out of the $\eta = K\dot{\gamma}^{n-1} + m\tau_0$ regime that governs viscosity below the yield stress in Eq. (6). In contrast, the microstructural aging process that is inherent in Eqs. (8) and (9) allow the time-dependent models in Case 1 to develop reasonable flow conditions despite the higher $\tau_0$. No doubt the time-independent models could be made to perform better in Case 1 by further manipulation of the model coefficients; however, our approach was to use coefficients that could be set *a priori* based on data from the experiments and a plausible $m$ value from Section 6.

    We observe that the simplified theoretical front prediction from Eq. (18), the dashed line in the figures, is a good representation of Chanson's Type II flows (Case 1 and Case 2) up until $t^* \sim 3$, but diverges rapidly thereafter. Our 2D simulations consistently overpredict the experimental front propagation in the early stages for Cases 1 and 2, but show better agreement with experiments than the simplified theory for $t^* > 4$. However, for Case 3 (Type I flow), the simplified theory is relatively poor, while the 2D simulations have good agreement up until $t^* \sim 3$, and then show significant underprediction of the experiments. As noted by Chanson et al. (2006), the Case 3 (Type I) experiments are at higher Reynolds numbers that, although

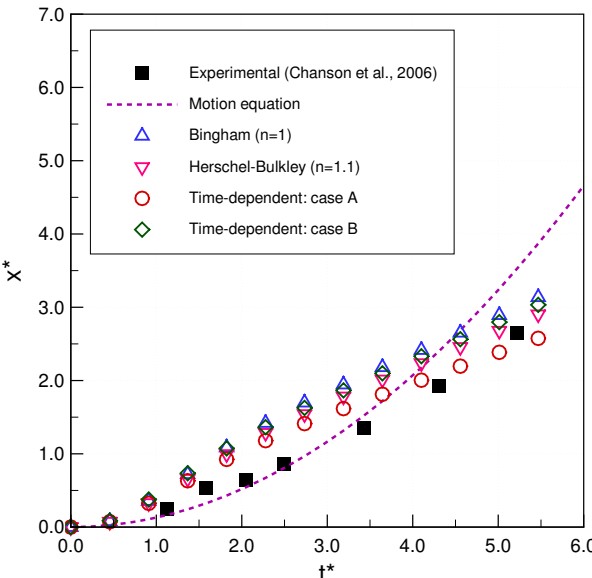

**Figure 8.** Thixotropic avalanche Case 2: comparison of numerical results and experimental data for non-dimensional front displacement ($x^*$) as a function of non-dimensional simulation time ($t^*$)

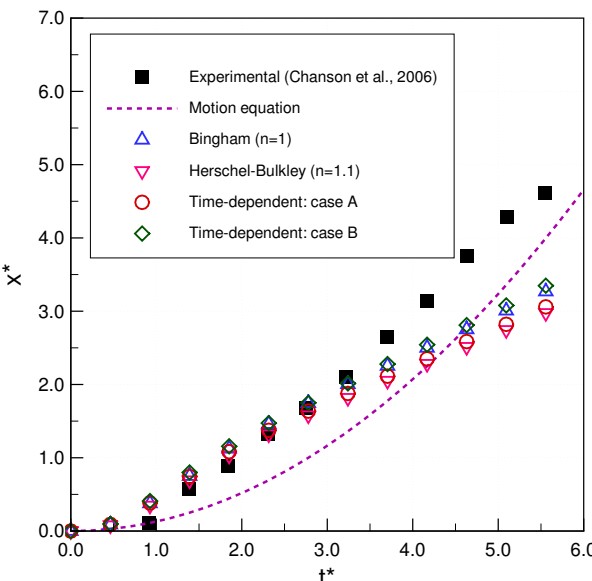

**Figure 9.** Thixotropic avalanche Case 3: comparison of numerical results and experimental data for non-dimensional front displacement ($x^*$) as a function of non-dimensional simulation time ($t^*$)

theoretically laminar, may be transitioning to weakly turbulent. Because simplified theory of Eq. (18) is derived by neglecting inertia, it is not surprising that its performance degrades with increasing Reynolds number.

Although the simulations results have reasonable global agreement with experiments, on closer examination it can be seen that the 2D simulations predict a front movement that is initially too rapid in Type II flows (Case 1 and 2), and at longer times is too slow for Type I flows (Case 3). The challenge, of course, is the model error is integrative: if $\lambda$ is wrong at a given time, then the $d\lambda/dt$ will be wrong as well and the frontal position error will accumulate. Thus, an important issue for the time-dependent model appears to be selecting the appropriate values of $\{\lambda_0, \alpha, \omega\}$ that are consistent with experimentally-determined values of $\{\eta_0, \tau_0, \tau_c, n, T_0\}$. Although the more parsimonious time-independent model (with fewer parameters) performs reasonably well for our Case 2 and 3, it performs poorly in Case 1 and so should only be used with caution and careful calibration.

The above observations lead to a conclusion that the accelerative behaviors in the simulations and experiments are not well matched. The problem is shown most clearly in Fig. 7 for Case 1, where the experiments initially follow the acceleration implied by Eq. (18), but diverge with an inflection point and deceleration occurring somewhere near $t^* \sim 4$. In contrast, the models initially show a more rapid acceleration and an inflection point to deceleration at $t^* \sim 1$. Interestingly, the simulated front locations in Cases 1 and 2 are not unreasonable predictions for $t^* > 4$, but they get there along slightly different paths than the experiments. The Case 3 (Type I) models show different behaviors: they perform quite well for $t^* \leq 3$ and then show deceleration at the same time as the experiment appears to be accelerating. Unfortunately, the experiments of Chanson et al. (2006) did not extend beyond $t^* \sim 6.5$, so it is impossible to know whether the experiments are showing an inflection point to deceleration at $t^* \sim 5$, but it seems likely given the results of the Case 1 and 2 studies. If there is an inflection point for Case 3, then it would appear that the consistent problem with the models is getting the correct transition from frontal acceleration to deceleration. To date, our experiments have not shown that we can significantly alter the model acceleration inflection points by altering parameters, which may indicate that there is a need to further consider the fundamental forms of the Coussot and Papanastasiou models when used for thixotropic flows. An alternative explanation may be that there are three-dimensional controls on the front propagation in the experiment that cannot be represented in the present 2D model.

## 8    Subaqueous Debris Flows:

In general, subaqueous debris flows are heterogenous gravity flows where the interaction of the overlying water with the downslope flow of the debris has a significant effect on momentum. Such flows are expected to be qualitatively similar to the subaqueous mud flow examined in the laboratory by Haza et al. (2013). We conducted simulations matching the Haza et al. (2013) experimental cases with the largest density difference between the water and mud. These conditions provide the largest effective negative buoyancy for the debris and minimize effect of turbidity. The selected cases are $35\%$ and $30\%$ KCC (Kaolin Clay Content). The schematic design is shown in Fig. 10, with dimensions provided in Table 4. The gravitational constant for all simulations is $g = 9.81 \text{ m} \cdot \text{s}^{-2}$.

The simulation uses $340 \times 100$ rectangular cells. The no-slip wall boundary condition is applied to the bottom boundary. The computational domain is rotated so the $x$-axis is parallel to the slope, which allows the bottom to be represented as a straight

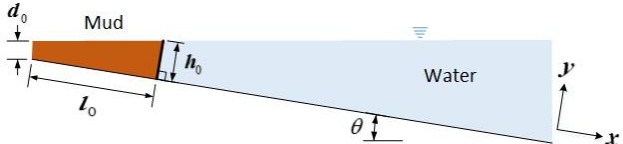

**Figure 10.** Submarine landslide

**Table 4.** Dimensions for simulations to match experiments by Haza et al. (2013).

| Term | Value |
| --- | --- |
| Angle of a slope ($\theta$, °) | 3 |
| Height of mud at a gate ($h_0$, cm) | 20.64 |
| Height of mud at the end ($d_0$, cm) | 15.40 |
| Length of mud ($l_0$, cm) | 100.0 |

surface without using cut grid cells or unstructured grids. This rotation also provides convenience in measuring the variables normal to the slope (e.g. front distance, and water/mud thicknesses at the front.) These simulations include three fluids: mud, water, and air. The density of mud for each case is shown in Table 6, and the densities of water and air are $1000.0 \, \mathrm{kg \cdot m^{-3}}$ and $1.0 \, \mathrm{kg \cdot m^{-3}}$, respectively.

The parameters for the time-independent fluid model from Haza et al. (2013) are shown in Table 5. For all simulations, $m = 100$ for the exponential smoothing parameter is used based on results from Section 6, above. The parameters for the time-dependent fluid model are estimated from Method B in Section 4 and are shown in Table 6. The experiments did not report a rest time, so $T_0$ was set at a small positive value that provided a reasonable match the experiments. The analytical relationships between the shear stress and the rate of strain for the time-independent and the time-dependent fluid models are

shown in Fig. 11 for Case 1 and Fig. 12 for Case 2.

Figure 13 provides a reference for measurements used to compare the model and experiments. These include the height of head-flow ($H$), the water depth at the front of head-flow ($D$), the run-out distance from the initial position ($L$), and the flow-front velocity ($U$). Figure 14 shows evolution of the zero level sets for water ($\phi_2$), which provides the continuous line separating the water from both the debris and the air. Figures 15 and 16 show the evolution of the run-out distance ($L$) for Case

1 and Case 2, respectively. It can be seen that both time-independent and time-dependent models are reasonable approximations of the limited experimental data. Both types of models appear to underestimate the initial run-out and slightly overestimate later times.

Figures 17 and 18 show a comparison of the height of head-flow ($H$) and water depth at the front ($D$) for simulations and experiments. Again, within the limited available experimental data, both time-independent and time-dependent model provide

reasonable agreement. Figures 19 and 20 show similar agreement for the front velocities, although the experimental data is insufficient to validate the wave-like oscillation of the velocity in the simulations.

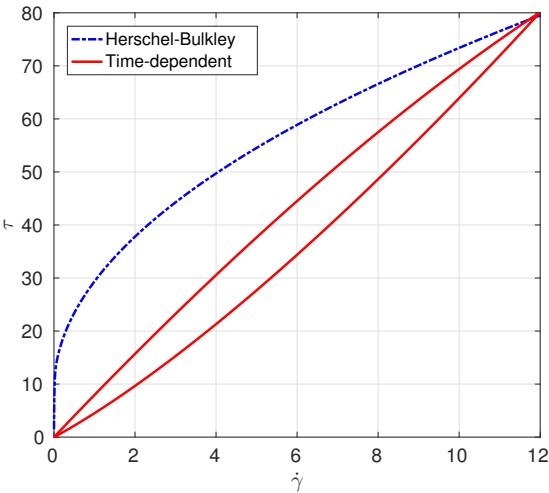

**Figure 11.** Subaqueous debris Case 1: shear stress (Pa) and rate of strain (s$^{-1}$)

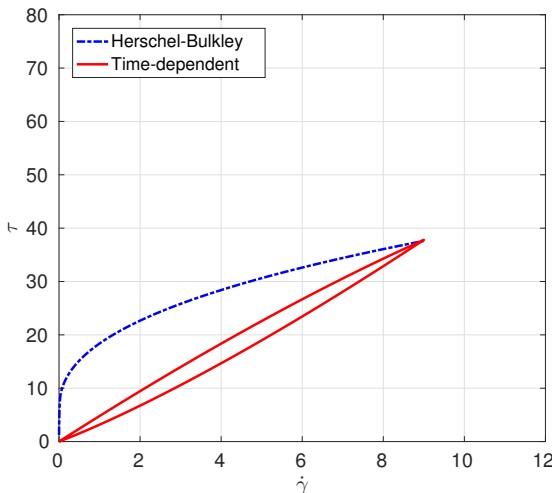

**Figure 12.** Subaqueous debris Case 2: shear stress (Pa) and rate of strain (s$^{-1}$). Axes scalings are identical to Fig. 11 for comparison purposes.

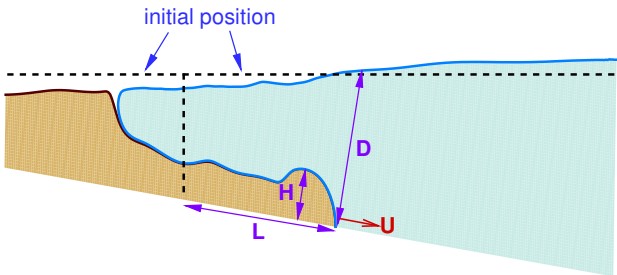

**Figure 13.** Run-out and head-flow

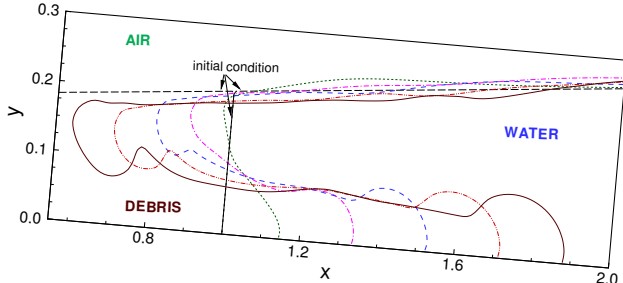

**Figure 14.** Profiles of debris and water ($\phi_2$: water)

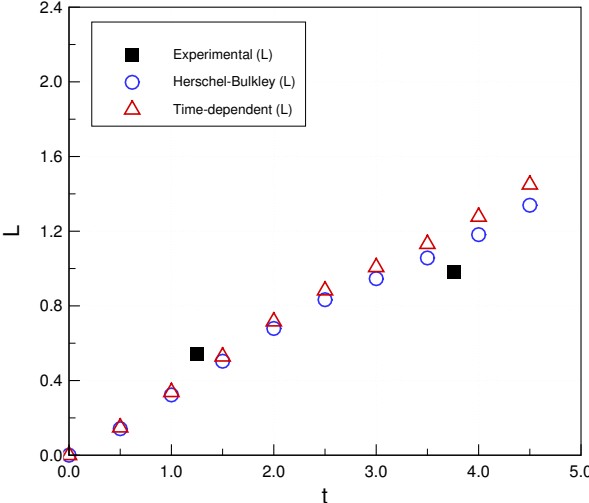

**Figure 15.** Subaqueous debris Case 1: run-out distance ($L$, m) as a function of simulation time ($t$, s)

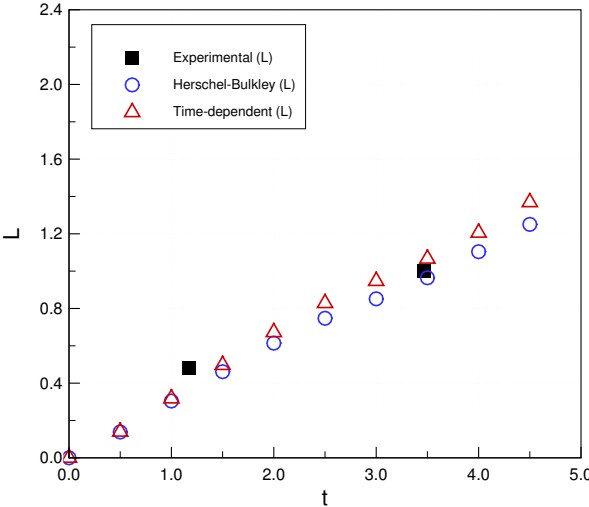

**Figure 16.** Subaqueous debris Case 2: run-out distance ($L$, m) as a function of simulation time ($t$, s)

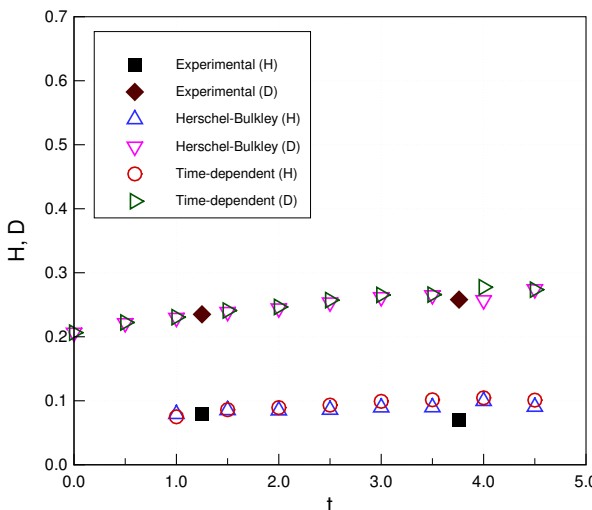

**Figure 17.** Subaqueous debris Case 1: height of head-flow ($H$, m) and water depth ($D$, m) as a function of simulation time ($t$, s)

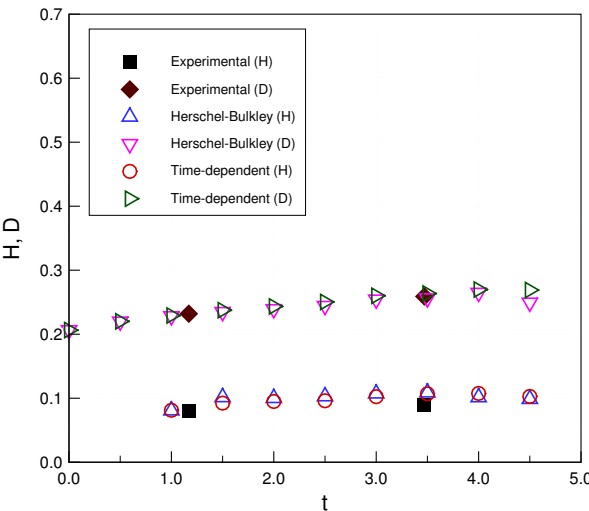

**Figure 18.** Subaqueous debris Case 2: height of head-flow ($H$, m) and water depth ($D$, m) as a function of simulation time ($t$, s)

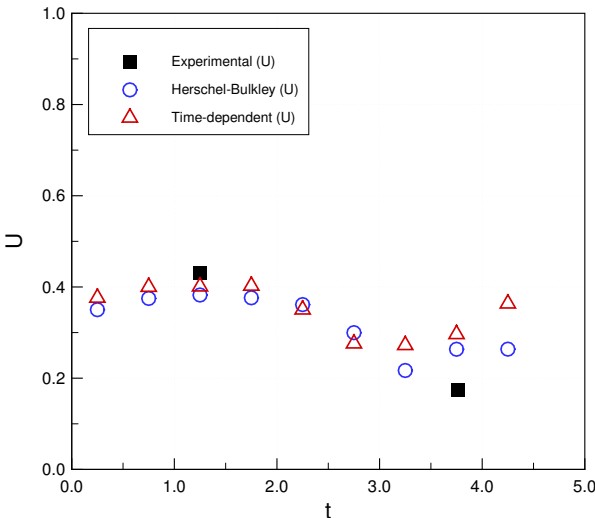

**Figure 19.** Subaqueous debris Case 1: flow-front velocity ($U$, m/s) as a function of simulation time ($t$, s).

**Table 5.** Parameters of the time-independent fluid model

| Term | Case 1 | Case 2 |
|---|---|---|
| Herschel-Bulkley index ($n$) | 0.5 | 0.42 |
| Yield stress ($\tau_0$, Pa) | 9.0 | 5.7 |
| Consistency parameter ($K$, $Pa \cdot s^n$) | 20.36 | 12.68 |

**Table 6.** Parameters of the time-dependent fluid model

| Term | Case 1 | Case 2 |
|---|---|---|
| Density ($\rho$, $kg/m^3$) | 1266.0 | 1236.0 |
| Asymptotic viscosity ($\eta_0$, $Pa \cdot s$) | 3.12 | 2.1 |
| Herschel-Bulkley index ($n$) | 0.5 | 0.42 |
| Flow index ($\omega$) | 1.0 | 1.0 |
| Characteristic time ($T_0$, s) | 10.0 | 10.0 |
| Material parameter ($\alpha$) | 1.0E-5 | 1.0E-5 |
| Microstructural parameter ($\lambda_0$) | 0.1 | 0.1 |

These results indicate the multi-material level-set model is capable of representing the key features in a subaqueous debris flow. For this flow, the use of the simpler time-independent viscosity model seems justified, although this is likely a function of the experimental conditions. An important limitation of the tested subaqueous debris flows is that they do not have the "restructuralization" in the downstream flow or the strongly jammed initial structure seen in the experiments of Chanson et al. (2006)

## 9    Discussion and Conclusions

This work shows that a multiphase flow solver using a multi-material level-set method with yield-stress models of non-Newtonian viscosity provides a means for numerical approximation of avalanches and subaqueous debris flows. This simulation approach was tested with both time-independent (Herschel-Bulkley, Papanastasiou, Bingham plastic) and time-dependent (thixotropic Coussot) models of viscosity, which are implemented using continuum mechanics solutions for multiple fluids. A key problem is that the Coussot model requires more parameters than the time-independent fluid models, but available experimental data are insufficient to definitively set parameter values. To resolve this issue, two different approaches were used to evaluating the Coussot parameters. Overall, the numerical results showed reasonable agreement with prior experimental data.

Although stress-strain relationships indicate the time-dependent approach provides the hysteresis that is desirable in a debris flow model, in comparisons with experimental data the time-dependent Coussot model provides a clear advantage for only for a single case – where the Herschel-Bulkley and Bingham plastic models erroneously predicted near-zero flow. Nevertheless,

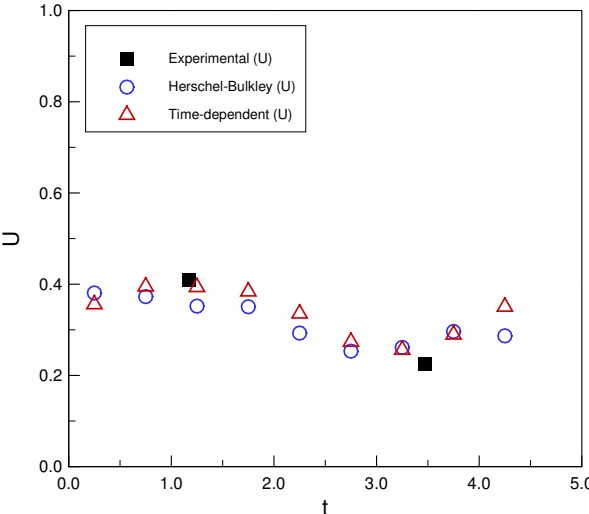

**Figure 20.** Subaqueous debris Case 2: flow-front velocity ($U$, m/s) as a function of simulation time ($t$, s).

much of the complexity in real-world behavior for debris mixtures is due to interactions across spatial scales for heterogeneous mixtures, which leads to significantly different stress/strain relationships during structural breakdown and restructuralization that should require a time-dependent model. Unfortunately, for experimental simplicity most researchers expend significant effort to create a homogeneous mixture as an initial condition for a debris flow, and the extent to which the structural breakdown

results in temporary heterogeneous scales is unknown. Existing laboratory data do not provide sufficiently detailed insight into the processes controlling destruction of jamming or the restructuralization of the flow, which leaves significant uncertainty in specification of the correct parameters.

     The time-independent viscosity-stress relationships that are often used for non-Newtonian flow models of natural hazards are a subset of possible viscosity-stress models. We believe that more complex models may be necessary for real-world heteroge-

neous mixtures that include hysteresis in the stress/strain relationship as microstructure evolves with time. In particular, where a fluid at rest has a strongly jammed structure or undergoes restructuralization as the flow slows, the time-independent Bingham plastic and Herschel-Bulkley models will likely be inadequate. Unfortunately, the processes by which the initial jamming is locally overcome, and the processes through which the structure is recovered, are both poorly understood. For time-dependent thixotropic models to be useful in modeling real-world avalanches and debris flows, there is a need for a consistent approach

to defining the initial jamming ($\lambda_0$), the characteristic time of aging ($T_0$), and the asymptotic shear viscosity ($\eta_0$), along with the material parameters $\omega$ and $\alpha$ for real-world systems. As yet, these parameters are not well-defined for either simple laboratory models or complex real-world flows. To improve our understanding of the thixotropic model, there is a need for a comprehensive sensitivity analysis of these five driving parameters for the expected scales of real-world systems (which are as yet unknown). Furthermore, with or without the thixotropic model, there is clearly a need for (1) more detailed experimental

measurements during flow initiation and restructuralization, and (2) a better understanding of the relationship between measurable microstructure parameters and the effective stress-strain relationship. The present work shows that a time-dependent (thixotropic) viscosity model may be an effective proxy for representing the inception and stalling of an avalanche or debris flow, but much work remains to be done before real-world natural hazards can be modeled in this manner.

5 *Acknowledgements.* The authors acknowledge the Texas Advanced Computing Center (TACC) at The University of Texas at Austin for providing HPC, visualization, database, or grid resources that have contributed to the research results reported within this paper. URL: http://www.tacc.utexas.edu. Publication support was provided by the Center for Water and the Environment at UT Austin and the Carl Ernest and Mattie Ann Muldrow Reistle Jr. Centennial Fellowship in Engineering.

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
