# Peer review of "Comparing Thixotropic and Herschel-Bulkley Parameterizations for Continuum Models of Avalanches and Subaqueous Debris Flows"

_Natural Hazards and Earth System Sciences, 2017_

## Referee Comment (RC1) · Anonymous Referee #1 · 1 Sep 2017

Manuscript ID nhess-2017-258 entitled "Comparing Thixotropic and Herschel-Bulkley Models for Avalanches and Subaqueous Debris Flows" which the authors submitted to the NHESS has been reviewed. The manuscript contains very interesting information, and very educative for debris flow rheologist, particularly for understanding of the rheology of clay with respect to the debris flow mobility in solid-liquid transition.

Accepted, but minor revision is recommended. Recommended comments are as follows:

In title, the terms of "avalanches and subaqueous debris flows" were used in the text; they are also shown in Session 6 and Session 7. In Discussion, the authors mentioned

[Figure]

"mudslides". Locat and Lee (2002) at Canadian Geotechnical Journal had presented the landside classifications: submarine landslides can be classified as five different types, such as spread, slide, flow, topple, and fall. It can be recommended that the authors should be defined/explain them in the manuscript: what is the difference from avalanche and debris flow. In Page 10, line 17-18, the avalanches were explained, but not for subaqueous debris flow.

In Table 1, 2, and 5, the yield stresses used in the text are very small, which are ranged from 5 to 30 Pa; it seems to me that the materials considered in this paper are in the stage between fully fluidized muds as a clay suspension and at liquid condition. As noted in the text, Chanson et al. (2006) used the bentonite clays, with volumetric concentration of solid ranging between 10 and 17%, that are mixed with fresh water. In reality, actual subaqueous debris flows are run with large-sized particles during debris flow motion. They are having a large yield stress value, which can be ranged from 1000 to 5000 Pa, even for mud-rich materials. Please explain the role of clays contained avalanche/debris flow/mudslides in subaqueous environment and how they influence upon the landslide motion.

No conclusions in this paper?

Recommendation:

Page 2, line 6, 19 (moller et al.) should be checked with page 7, line 16. Page 2, line 23, yield stress vs page 23, line 2 yield-stress Page 6, line 18 ODE (9): ODE (Eq. 9)? Page 7, lineg 7, 8: Session 6, Session 7? Page 12, Fig. 3: the shape of landslide is triangular? Why not for parabolic shape? Any reason? Page 15, Fig. 7: x-axis of nondimensional time (t*) is the time scale for the landslide initiation or landslide motion or debris flow propagation?

---

## Referee Comment (RC2) · Anonymous Referee #2 · 19 Sep 2017

General remarks:

This paper presents an interesting work conducted on the rheological characterization of natural heterogeneous mixtures (e.g. debris flows, avalanches, etc.). The paper is clearly written and the figures are all useful. Most of the information is very useful for viscous fluids rheology. The paper should be accepted with minor revision. Some specific comments:

- Authors should clarify the different terms used for the definition of the flow-like phenomena they are discussing. They should insert a synthesis figure showing the rheological characteristics of natural flow-like phenomenon discussed within this work (i.e.

debris flows, avalanches, etc.). They should insist on the main differences between those fluids in terms of viscosity, mono/biphasic fluid, grain-size distribution, etc.). They should also mentioned somewhere that lahars are considered as very specific viscous fluid in most of those classifications;

- Authors do not discuss some key elements concerning debris flows: (1) the triggering can be a fluidization of deposits within the channel or laying on connected side slopes (solid to fluid), but also can be a enrichment of a "classical" flood with solid material during the runout (fluid to solid); (2) the rheology of a single debris flow event can vary during its runout due to entrainment processes;

- Authors should explain somewhere the influence of clays on the flow motion and how it varies according this clay content;

- A comparison between their results and observations of real study case is missing. They could insert a simple table with the main rheological and morphological characteristics of their experiments and some characteristics of other case studies;

- A sensitivity analysis could be discusses somewhere in the discussion part where authors could identify which input data has the most influence on the output data;

- Where's the final conclusion?

Specific remarks:

Page 14, Figs 5 & 6: Why the scale of X and Y-axis of both graphs are different? It could mislead the readers;

Page 19, Figs 11 & 12: Same remark as above.

---

## Author Comment (AC3) · 9 Nov 2017

Dear Editor,

We would like to modify the title of our paper if possible:

From: "Comparing Thixotropic and Herschel-Bulkley Models for Avalanches and Subaqueous Debris Flows"

To: "Comparing Thixotropic and Herschel-Bulkley Parameterizations for Continuum Models of Avalanches and Subaqueous Debris Flows"

We believe the modified one would represent the main focus of our paper better.

---

## Author Response (AR1)

**Answers for the comments by the Editor**

**(1) The authors should certainly state again, though they build their work on a long history of use of non-Newtonian rheology for debris slurries, that material resistance to the flow, expressed by the rheological law, is only one aspect of the physics of these events. Debris flow slurries can be simulated only by rheological models or by fluid-solid interaction models and that according to the type of events, the objective of the modelling or the scale of analyses, both approaches may be valid. In this sense, I would suggest that the, still somehow provocative, manuscript by Iverson (2003, 3rd DFHM Conference) on the debris flow rheology myth might be cited.**

We have added a paragraph explaining Iverson's approach in the introduction on pg 3, lines 4-12 as:

–" Indeed, Iverson (2003) has referred to the rheological approach to debris flows as a "myth" and argued for its replacement with mixture models using separate solid-fluid components. However, their argument remains contentious and it is not clear that the present state of mixture models is substantially less mythical than application of a rheological model when considering heterogeneous mixtures over a wide range of scales. Given that debris flow covers such diverse phenomena and complex physics, it seems likely the "correct" model for the foreseeable future will be the model that best fits a specific event, experiment, or flow type of interest. In the absence of research that definitively solves the conundrum of debris flow, we follow the long history of using rheological models as a proxy. Such models are parsimonious in the number of coefficients and are effectively agnostic to the inherent uncertainties of fluid-solid distributions and interactions. "

**(2) references to more or less recent works on either experimental analyses of different material rheologies of slurries or based on real observations might reinforce the broad overview (e.g. state of the art) carried out by the authors on which rheology to choose.**

We have added some references in the introduction on pg 2, lines 14-19 as:

–"in any model attempting to directly represent fluid-solid structural interactions.

Large-scale natural hazard flows have been widely investigated with field observations, small-scale laboratory experiments, and numerical models. A common observation is that the complexity of the material composition and the effective rheological characteristics play important roles in material movement (Malet et al.,2003; Bisantino et al., 2010; Jeong, 2014; de Haas et al., 2015). "

**Answers for the comments by Referee 1**

**(1) The authors should be defined/explain them (classification of submarine landslides) in the manuscript: What is the difference from avalanche and debris flow. In Page 10, line 17-18, the avalanches were explained, but not for subaqueous debris flow.**

We have added the definitions in the introduction on page 2, lines 6-8 as:

–"Avalanches (e.g. snow, rock) are typically considered dry granular flows, whereas debris flows are liquid/solid mixtures where the solids are a dominant forcing, which can be contrasted to flood flows where sediment solids play a secondary role (Iverson,1997)."

We have also tried to clarify how we are treating these disparate flows similarly on page 2, lines 29-31

– "Herein, we do not seek to distinguish between the differing physics of these various complex flows, but focus on advancing the use of non-Newtonian viscosity models as a proxy for their general behavior. For simplicity in exposition, we will use the term "debris flow" to refer to any real-world mixture modeled as a continuum fluid using a non-Newtonian model."

**(2) In reality, actual subaqueous debris flows are run with large-sized particles during debris flow motion.**

We have combined the answer to this comment with the answers to comment (3), see below.

**(3) Please explain the role of clays contained avalanche/debris flow/mudslides in subaqueous environment and how they influence upon the landslide motion.**

We have combined the answer to this comment with the answer to comment (2). These comments are also similar to the comments (1) and (2) of Referee 2. The reviewers have pointed out a problem in the way we presented the paper in the introduction. Comments (1) and (2) involve the specific physics of debris flows associated with real-world particle sizes and physics of clays. Our original paper was not sufficiently clear in the abstract and introduction: we are not intending to look at the physics of these flows, but instead are focused on the way in which previously-used models might be improved by adoption of thixotropy in a non-Newtonian approach where viscosity is a proxy for all the complex physics. As such, we do not believe it is appropriate to get into a detailed discussion of the physics that are only approximated in their effects by a non-Newtonian model. We have significantly rewritten the abstract and introduction to make this clearer. This includes pages 2-3, lines 1-14, 19-31 (page 2) and lines 1-3 (page 3), along with minor clarifications throughout page 3. In particular, see page 2, lines 26-30, where we provide the reader with references for the physics and place our work in a clearer context:

– "We take these issues as motivational for the present work and refer the reader to the recent review of Delannay et al. (2017) for further insight on granular flows and Shanmugam (2015) for heterogenous flows. The fundamentals physics of such flows is presented in Iverson (1997). Herein, we do not seek to distinguish between the differing physics of these various complex flows, but focus on advancing the use of non-Newtonian viscosity models as a proxy for their general behavior."

Thank you for these comments as they helped us significantly improve the introduction to the paper.

**(4) No conclusions in this paper?**

Our conclusions were integrated in the prior Discussion section. We have modified the name to "Discussion and Conclusions" as it contains both the summary of the results and observations as to the impact of the two sets of results taken together. Note that additions to the closing paragraph have been provided to provide better insight into how this work could be extended in the future. See pages 26 (line 13) - 27 (line 4):

– "For time-dependent thixotropic models to be useful in modeling real-world avalanches and debris flows, there is a need for a consistent approach to defining the initial jamming ($\lambda_0$), the characteristic time of aging ($T_0$), and the asymptotic shear viscosity ($\eta_0$), along with the material parameters $\omega$ and $\alpha$ for real-world systems. As yet, these parameters are not well-defined for either simple laboratory models or complex real-world flows. To improve our understanding of the thixotropic model, there is a need for a comprehensive sensitivity analysis of these five driving parameters for the expected scales of real-world systems (which are as yet unknown). Furthermore, with or without the thixotropic model, there is clearly a need for (1) more detailed experimental measurements during flow initiation and restructuralization, and (2) a better understanding of the relationship between measurable microstructure parameters and the effective stress-strain relationship. The present work shows that a time-dependent (thixotropic) viscosity model may be an effective proxy for representing the inception and stalling of an avalanche or debris flow, but much work remains to be done before real-world natural hazards can be modeled in this manner."

**(5) Page 2, line 6, 19 (moller et al.) should be checked with page 7, line 16.**

We have made citations to "Møller et al. (2006)" consistent with the spelling in that paper. Note that the citation to "Moller et al. (2009)" is also consistent with the spelling in that paper as the latter did not include the ø in the author's name.

**(6) Page 2, line 23, yield stress vs page 23, line 2 yield-stress**

We have corrected the paper for consistent grammar throughout. Where we use "yield-stress fluid" we include the dash as "yield" modifies "stress" rather than "fluid" such that "yield-stress" is a compound adjective. However, where we write "yield stress" with stress as a noun, there is no need for a dash.

**(7) Page 6, line 18 ODE (9): ODE (Eq. 9)?**

Modified to "the ordinary differential equation presented as Eq. (9)".

**(8) Page 7, lines 7, 8: Session 6 , Session 7?**

Modified the symbol "§" to "Section" throughout the paper.

**(9) Page 12, Fig. 3: the shape of landslide is triangular? Why not for parabolic shape? Any reason?**

The shape was chosen to match initial conditions of prior experiments in the literature. In the experiments, a retaining gate is opened and the avalanche starts to flow. During the flow the shape tends toward parabolic.

**(10) Page 15, Fig. 7: x-axis of nondimensional time (t*) is the time scale for the landslide initiation or landslide motion or debris flow propagation?**

The time scale denotes the simulation time ($t$) after gate opening. It is non-dimensionalized by the gravity ($g$) and the initial height normal to the bottom ($d_0$). In other words, it is the time of flow motion (landslide or debris flow). To make this clear, we added some description for this to the manuscript, see page 15, line 7:

– "... the non-dimensionalized front location and **simulation** time **after gate opening** are..."

We also added "simulation time" to captions of figures 7, 8, and 9.

**Answers for the comments by Referee 2**

(1) **Authors should clarify the different terms used for the definition of the flow-like phenomena they are discussing. They should insert a synthesis figure showing the rheological characteristics of natural flow-like phenomenon discussed within this work. They should insist on the main difference between those fluids in terms of viscosity, mono/biphasic fluid, grain-size distribution, etc. They should also mentioned somewhere that lahars are considered as very specific viscous fluid in most of thosed classifications. Authors do not discuss some key elements concerning debris flows: (1) the triggering can be a fluidization of deposits within the channel or laying on connected side slopes, but also can be a enrichment of a 'classical' flood with solid material during the runout, (2) the rheology of a single debris flow event can vary during its runout due to entrainment processes.**

We have combined the answer to this comment with the answers for comment (2) below.

(2) **Authors should explain somewhere the influence of clays on the flow motion and how it varies according this clay content.**

We have combined the answer to this comment with the answer for comment (1). These comments are similar to the comments (2) and (3) of Referee 1 in that they seek further details on the physics of real-world flows. In retrospect, we believe the original paper did not place our work carefully in context of prior work where non-Newtonian models are used as proxy for the macroscopic behavior of real-world flows. We have substantially rewritten the abstract and introduction to make it clearer that we are only aiming to understand the different behaviors that can be simulated by using a thixotropic model rather than a Herschel-Bulkley model for representing debris flows and avalanches. As we are building on a long history of using non-Newtonian physics as a proxy for these flows, we do not believe that it is appropriate to delve into the details of the rheology. However, these comments have helped us to see that we were deficient in our presentation of the background and motivation, which left the reader confused as to our purpose and scope. We have substantially expanded our introduction and included further references to review papers in the literature where more details on flow physics can be found.

In particular, the Referee's questions are addressed in new text, page 2, lines 1-14 and 19-31, and page 3, lines 1-3 regarding general characteristics and definitions of different flows, along with citations of review papers for physics. The Reviewer will find their ideas directly addressed in this new text.

Also, see new text page 6, lines 5-11 regarding clays in laboratory flows vs. real debris flows:

– "Thixotropic flows modeled at the laboratory scale typically use clays (e.g. Bentonite, Kaolin) to create the microstructure controlling non-Newtonian behavior (citation). Preparation of a homogenous clay suspension for such experiments is a demanding task, the details of which can be found in (citations). Unfortunately, we cannot expect the structure of a heterogenous large-scale debris flow to mimic the flow scales, yield stresses, and parameters for a homogeneous thixotropic laboratory flow. However, lacking data from a large-scale debris flow that could be adequately used for model comparisons, herein we take a first step by analyzing how thixotropic models compare to time-independent models for laboratory-scale flows."

(3) **A comparison between their results and observations of real study case is missing. They could insert a simple table with the main rheological and morphological characteristics of their experiments and some characteristics of other case studies.**

We believe our comparisons to laboratory experiments are as far as can be reasonably achieved in the present work. We have pointed out the challenge of validation to field experiments.

New text, page. 6, lines 12-17:

– "Validating the use of a non-Newtonian model to represent a real-world debris flow presents challenges on two levels: first, does the model correctly represent a non-Newtonian flow? Second, does the non-Newtonian flow (when parameterized) represent a real-world debris flow? To date, successful non-Newtonian models of real-world flows have been parameterized using a time-independent approach, which limits the ability of the model to represent the transition phases, i.e. flow initiation and stalling (citations provided). Unfortunately, data on transition phases for real-world flows is lacking, and is severely limited even for laboratory-scale flows."

**(4) A sensitivity analysis could be discussed somewhere in the discussion part where authors could identify which input data has the most influence on the output data.**

We agree that a sensitivity analyses would be useful; however, with 5 parameters to test over essentially unknown ranges (due to the lack of sufficient experimental or field data) we believe this is beyond the reasonable scope for the present paper. We have added the following in the Discussion and Conclusions section to highlight this point.

New text, page 26 lines 17-19:

– "To improve our understanding of the thixotropic model, there is a need for a comprehensive sensitivity analysis of these five driving parameters for the expected scales of real-world systems (which are as yet unknown)."

**(5) Where's the final conclusion?**

Our conclusions were integrated in the prior Discussion section. We have modified the name to "Discussion and Conclusions" as it contains both the summary of the results and observations as to the impact of the two sets of results taken together. We rewritten this section and provided a new final paragraph that provides greater insight into the future possibilities for this method.

New text, pages 26 (line 13) - 27 (line 4).

– "For time-dependent thixotropic models to be useful in modeling real-world avalanches and debris flows, there is a need for a consistent approach to defining the initial jamming ($\lambda_0$), the characteristic time of aging ($T_0$), and the asymptotic shear viscosity ($\eta_0$), along with the material parameters $\omega$ and $\alpha$ for real-world systems. As yet, these parameters are not well-defined for either simple laboratory models or complex real-world flows. To improve our understanding of the thixotropic model, there is a need for a comprehensive sensitivity analysis of these five driving parameters for the expected scales of real-world systems (which are as yet unknown). Furthermore, with or without the thixotropic model, there is clearly a need for (1) more detailed experimental measurements during flow initiation and restructuralization, and (2) a better understanding of the relationship between measurable microstructure parameters and the effective stress-strain relationship. The present work shows that a time-dependent (thixotropic) viscosity model may be an effective proxy for representing the inception and stalling of an avalanche or debris flow, but much work remains to be done before real-world natural hazards can be modeled in this manner."

**(6) Page 14, Figs. 5 & 6 ; Page 19, Figs. 11 & 12 : why the scale of X and Y-axis of both graphs are different? It could mislead the readers.**

The $\tau$ y-axes of figures 4, 5 and 6 use the same scale so that the comparisons can between figures can be readily made. However, the $x$ axes have individual scales for clarity due to the large change in range. We agree this can be confusing, so we've added the following note to the captions of Figs. 4, 5 and 6:

– "The $\tau$ axis is scaled for comparison with Figs. 5 and 6 while the $\dot{\gamma}$ axis has an individual scale for clarity."

Figure 12 now has exactly the same x and y axes as Figure 11, which is now noted in the caption of Figure 12 as:

[revised manuscript text omitted]